# Determination of Efficacy of Single and Double 4.7 mg Deslorelin Acetate Implant on the Reproductive Activity of Female Pond Sliders (*Trachemys scripta*)

**DOI:** 10.3390/ani11030660

**Published:** 2021-03-02

**Authors:** Edoardo Bardi, Martina Manfredi, Raffaella Capitelli, Emanuele Lubian, Alessandro Vetere, Alessandro Montani, Tommaso Bertoni, Elisa Talon, Gabriele Ratti, Stefano Romussi

**Affiliations:** 1Department of Veterinary Medicine, University of Milan, via dell’Università 6, 26900 Lodi, Italy; martina.manfredi@unimi.it (M.M.); gabriele.ratti@unimi.it (G.R.); stefano.romussi@unimi.it (S.R.); 2Csv-Labvet, via J. Kennedy 10, 23873 Missaglia, Italy; capitelli@libero.it (R.C.); elisa.talon@gmail.com (E.T.); 3Veterinary Clinical and Husbandry Centre (CCVZS), University of Milan, via dell’Università 6, 26900 Lodi, Italy; emanuele.lubian@hotmail.com; 4Clinica Veterinaria Modena Sud, Piazza dei Tintori 1, 41057 Spilamberto, Italy; alessandro.vetere88@gmail.com; 5Faculty of Veterinary Medicine, University of Teramo, Località Piano d’Accio, 64100 Teramo, Italy; alemontanivet@gmail.com; 6MySpace Lab, Department of Clinical Neuroscience, University of Lausanne, rue du Bugnon 21, 1011 Lausanne, Switzerland; tommaso.bertoni90@gmail.com

**Keywords:** contraception, deslorelin acetate, GnRH agonist, pond slider, reptile, *Trachemys scripta*, turtle

## Abstract

**Simple Summary:**

North American pond sliders (*Trachemys scripta*) are invasive alien species that, following the release of pet specimens in the wild over the decades, have established breeding populations outside their native habitat, with detrimental effects on local biodiversity. Their commerce and breeding have been recently banned in the European Union, and national programs are being introduced by Union Members to eradicate and control their presence. Among other means of population control, the use of long-acting drugs for chemically induced infertility could be a promising tool to prevent the reproduction of these animals. Efficacy of single and double intramuscular deslorelin acetate implants was investigated during a one-season follow-up: plasmatic sexual hormones concentration and morphometric evaluation of ovarian activity by computed tomography were performed about every 30 days to detect the differences between the control group (no implant), single-implant group, and double-implant group. Results show no inhibition of reproductive activity for the ongoing reproductive season, but suggest possible infertility starting from the following season.

**Abstract:**

The use of long-acting gonadotropin-releasing hormone (GnRH) agonists to suppress fertility has been poorly investigated in reptiles, and the few available studies show inconsistent results. The efficacy of single and double intramuscular 4.7 mg deslorelin acetate implants in captive pond sliders (*Trachemys scripta*) was investigated, with 20 animals divided into three groups: a single-implant group (6 animals), a double-implant group (6 animals), and a control group (no implant). During one reproductive season (March to October), plasmatic concentration of sexual hormones (estradiol, progesterone, and testosterone) and ovarian morphometric activity via computed tomography were monitored about every 30 days. A significative decrease in the number of phase II ovarian follicles was detected in the double-implant group compared with the control group, but no significant difference was noted in the number of phase III and phase IV follicles, egg production, and plasmatic concentration of sexual hormones. Results show that neither a single nor a double deslorelin acetate implant can successfully inhibit reproduction in female pond sliders during the ongoing season, but the lower number of phase II follicles in the double-implant group can possibly be associated with reduced fertility in the following seasons.

## 1. Introduction

American pond sliders (*Trachemys scripta*) are among the most common pet turtles all over the world since the 1970s, but as a consequence of their frequent release in the wild, they are considered invasive alien species in many countries. With well-established populations outside their native distribution [1,2,3,4], they pose a threat for native species [5,6,7,8,9,10,11] and possible risks for domestic animals and human health [12,13,14]. For these reasons, *T. scripta* is listed among the 100 most invasive species by the International Union for Conservation of Nature (IUCN) [15], and its breeding and commerce have been recently prohibited inside the European Union, with Regulations (EU) 1143/2014 and 1141/2016, leading to the necessity for means of population control. Synthetic gonadotropin-releasing hormone (GnRH) analogues are used to inhibit sexual hormone production, with deslorelin acetate (DA) being one of the most common GnRH agonists in veterinary practice. Its mechanism of action consists in a powerful stimulation of the pituitary gland to produce gonadotropins (flare-up phase) and consequent long-term downregulation of pituitary receptors for endogenous GnRH, leading to temporary suppression of fertility [16,17,18]. It is formulated as a controlled-released implant for subcutaneous administration in dogs and ferrets and is commercially available as 4.7 and 9.4 mg implants.

In avian species, DA implants are rarely employed, usually to medically reduce aggressivity or treat sex-related disorders such as excessive egg laying, with great variability in efficacy and duration [18,19,20,21,22,23]. In reptiles, GnRH agonists have been poorly investigated so far: in lizards, suppression of gonadic activity was reported only in iguanas [18,24,25,26]. Regarding chelonians, annual applications of DA implants reduced serum testosterone levels after the fourth treatment in a male *Chelonia mydas* [27]; DA was successfully used to treat chronic ovodeposition problems in a *Testudo graeca* up to 24 months after implantation [28]. In *T. scripta*, subcutaneous implantation showed no statistical differences in serum hormonal concentration or reproductive activity between case and control groups in both males and females [24,29], but males showed a transient increase in testosterone levels two weeks after implantation [29]. 

The purpose of the present work is to evaluate the effects of single and double intramuscular DA slow-release implants on ovarian and reproductive activity of mature female pond sliders, by serial measurements of hormonal plasmatic concentrations and morphometric evaluations of the ovaries. Even if no data are available regarding the elution rate of DA implants in reptiles, it was decided to use a double 4.7 mg implant instead of a single 9.4 mg implant based on the assumption that a double implant would have displayed a shorter onset and higher efficacy [21,30].

## 2. Materials and Methods

Twenty adult, clinically healthy female sliders (minimum plastron length 18 cm) [31] with a history of correct outdoor management were enrolled for the study. In mid-March 2019, each animal received a complete physical examination and a radiography to rule out egg retention from the previous reproductive season. At T0, the animals were individually marked with a nanochip (Nanochip for Exotic Animals, Therapet Bioforlife, Milano, Italy) in the left pectoral muscle, and each turtle received a full-body helical CT scan and a blood sampling to determine baseline values (see later) prior to DA implantation. Each animal was randomly assigned to group 1 (control group, 8 animals), 2 or 3 (6 animals each): animals assigned to group 2 received a single 4.7 mg DA implant (Suprelorin 4.7 mg, Virbac Italia, Milano, Italy) in the right pectoral muscle; animals assigned to group 3 received a double 4.7 mg DA implant in the right and left pectoral muscle. Time points were set every month, from T0 to T6, for a total of 7 time points (Figure 1). In between monthly evaluations, the animals were randomly housed in groups of 4 animals in outdoor seminatural conditions, with the possibility of laying eggs if needed; in each group, a healthy adult male was added to properly stimulate the females.

### 2.1. Ovarian Morphometric Evaluation

From T0 (prior to DA implantation) to T6, a full-body helical CT scan (GE BrightSpeed Elite 16 slice—slice thickness 1.25 mm, 0.938 pitch, 180 mAs, 120 kVp) was performed each month for ovarian morphometric evaluation. Sliders were placed in ventral recumbency on a flat surface during image acquisition. Images were reconstructed using both bone and soft tissue algorithms. The soft tissue algorithm series (window width: 400, window level: 40) was used for measurements on an open-source DICOM viewer (Horos v.3.3, https://www.horosproject.org, accessed on 1 April 2020). Total and differential count for follicles belonging to classes II, III, and IV of the Moll and Legler classification [3,31] was performed, and the presence and the number of eggs were recorded.

### 2.2. Plasmatic Hormones Concentration

From T0 (prior to DA implantation) to T6, a blood sample from the cervical plexus was obtained from each animal and stored in a lithium heparine test tube; plasma was immediately separated by centrifugation and frozen at −20 °C. The samples were processed no more than 15 days after collection. Estrogen (17β-estradiol, E_2_) and progesterone (P_3_) plasmatic concentration were measured for every time point. Testosterone (T) plasmatic concentration was measured at T0, T3, T4, T5, and T6. Fluorescence polarization immunoassay (FPIA, competitive enzyme immunoassay) technology of the Tosoh 600 Analyzer was used to evaluate plasma E_2_, P_3_, and T concentration. For E_2_, linearity was between 25 and 3000 pg/mL, while intra- and inter-assay variability were, respectively, 4.03% and 6%. For P_3_, linearity was between 0.1 and 40 ng/mL, while intra- and inter-assay variability were, respectively, 2.2% and 2.8%. For T, linearity was between 10 and 2200 ng/dL, while intra- and inter-assay variability were, respectively, 3.2% and 3.7%. 

### 2.3. Statistical Analysis

The number of follicles and the concentration of hormones were compared between experimental groups through two-way mixed-design ANOVAs, with group as a between-subjects variable and time as a within-subjects variable. After each ANOVA, assumption of normality was checked by running a Shapiro–Wilk test on the residuals. Statistical analyses were conducted through the R statistical software (version 4.0.2), using the afex package (https://CRAN.R-project.org/package=afex, accessed on 1 April 2020, version 0.28-0) for ANOVAs. Focus was put on the total number of follicles of stages II, III, and IV, and on the number of oviductal eggs: to evaluate the differences between groups over the whole observation period, two-way ANOVAs were run contrasting the control group to the single- or double-implant group. A significant group*time interaction would therefore indicate that the variation in time of the number of follicles is different between the compared groups, suggesting an effect determined by the DA implant(s).

Prior to running statistical analyses on hormone concentrations, a check was performed on the amount of missing or below-threshold values, which was 67% for E_2_, 12.1% for P_3_, and 32% for T. Estradiol values were therefore excluded from further analyses due to the amount of missing values. Moreover, analysis of the distribution of T and P_3_ found the latter to be affected by outliers, with 95% of the values below 1 ng/mL and 5% of the values going as high as 20 ng/mL. Values above 1 ng/mL were therefore excluded from further analyses.

## 3. Results

The first comparison, control vs. single-implant, yielded no significant results for the group*time interaction either for follicles (*p* = 0.85) or oviductal eggs (*p* = 0.77). As confirmed by visual inspection of the graphs (Figure 2), the single-implant group seems to start with a slightly lower number of follicles but decreases only slightly across the observation period in a manner similar to the control group. When running the same comparison against the double-implant group, instead, no significant result was found regarding the number of eggs (*p* = 0.23), but a significant group*time interaction (*F*(6.60) = 3.06, *p* = 0.011) was found regarding the total number of follicles. A Shapiro–Wilk test run on the residuals failed to reject the normality hypothesis (*p* = 0.065), confirming the validity of the analysis, and the result also holds when applying Greenhouse–Geisser and Huynh–Feldt corrections for departure from sphericity (*p* = 0.043). The two groups have matching values until T2, then the double-implant group decreases sharply, while the control group decreases only moderately, so the result is not driven by preexisting differences (Figure 2). To assess this effect quantitatively, the observation period was divided into two (pre: until T3 included; post: after T3), assuming T3 as a watershed between the first and further clutches of follicles [3,32], and the mean variation in the number of follicles was computed for each animal. A decrease in the number of follicles was observed for all groups (control: −4.90 ± 1.15, single-implant: −2.19 ± 5.47, double-implant: −11.2 ±1.59, mean ± SEM), but a decrease in the double-implant group was significantly larger than in the control group (Welch’s *t*-test, *t*(1, 9.12) = 3.21, *p* = 0.010). The decrease in the control group did not differ significantly from the decrease in the single-implant group (*p* = 0.65). 

To better investigate the source of the observed differences, a differential count for the three classes of follicles was performed (Figure 2; Table 1). The variations in the number of oviductal eggs are shown in Figure 3.

The same analyses as for the number of follicles were performed on T and P_3_ concentration values, comparing both the single- and the double-implant group with the control group. The residuals were normally distributed (all Shapiro–Wilk *p*-values above 0.23), and comparison yielded significant results, with all *p*-values above 0.22 (Figure 4). Since no statistical difference was noted between the groups, the values for progesterone and testosterone concentration are reported for the entire experimental population in Table 2. 

To investigate whether the measured concentrations of hormones could predict the total number of follicles, a correlation test between the average hormone concentration and the average number of phase II–IV follicles during the whole observation period was performed. The analysis evidenced a mild, nonsignificant correlation between progesterone levels and number of follicles (*R* = 0.465, *p* = 0.149). No correlation was found with testosterone levels (*R* = 0.247, *p* = 0.80).

## 4. Discussion

In the present study, and consistently with previously published works [24,29], both single and double 4.7 mg DA implants did not succeed in suppressing fertility in female pond sliders. However, some level of activity can be attributed to the double-implant group, since a significant difference in ovarian mass was noted with regard to the control group. Such difference in the total number of follicles was largely dependent on the variations of stage II (7–13 mm) follicles, with a significant decrease and slight increase at T6 in the double-implant group; stage III (14–20 mm, late vitellogenic) follicles were observed to progressively decrease in all three groups without significant differences, while the double-implant group showed a spike in the number of stage IV (>20 mm, preovulatory) follicles at T2 (65 days after implant administration), suggesting a stimulating effect of the DA implants on the hypothalamic–pituitary–gonadal axis [3,29,33]. Since stage II follicles are assumed to be related to the clutches of the following year [3], it can be hypothesized that the double-implant group could decrease or even suppress fertility starting from the second reproductive season after administration.

In all the three groups of the present study (excluding the aforementioned spike in stage IV follicles in the double-implant group), variations in ovarian mass were consistent with those described in wild *T. scripta* in Spain [3], with a decrease in total follicle number from May to July and a plateau from July to September. The same work observed an increase in total follicle number in November before winter hibernation, but this could not be confirmed by our results since our last evaluation was performed at the beginning of October. 

Physiologic hormonal pattern has not been described in pond sliders. In the control group of the present study, progesterone showed a peak at T1 followed by slight fluctuations, lowering at T2, T3, and T5, and rising at T4 and T6. Testosterone was stable between T0 and T1, dropped through T5, and rose again slightly at T6. These variations resulted similar to those described in other Emydidae and freshwater North American turtles [34,35,36], despite showing lower concentrations. Estradiol pattern could not be evaluated due to the unavailability of most values and consequent inadequacy for statistical analysis. In the present study, hormonal values were not useful to assess differences between the groups. No differences were found regarding the overall pattern of P_3_ and T, but some punctual differences were recorded. Regarding P_3_, a particularly low concentration was noted in the double-implant group at T1, preceding the above-described spike in stage IV follicles and thus probably correlated to the hypothesized flare-up phase. Testosterone pattern showed a decrease starting at T3, followed by a plateau phase in control and single-implant groups, while the double-implant group displayed an increase starting at T4 until T6. Such an increase was not correlated to any significant variation in follicular pattern neither at T4, T5, or T6; hypothetically, the increase could be interpreted as a preovulatory surge [34,35,36,37] but, lacking further evaluation in November to witness an increase in stage III and IV follicles, its biological meaning remains unclear. 

## 5. Conclusions

Neither single nor double deslorelin acetate implant was successful in suppressing gonadal activity and preventing reproduction in adult female pond sliders during a one-season follow-up. This result is consistent with that of previously published works [24,29], and morphometric evaluation of the ovaries by CT scan proved more useful than hormonal evaluation to monitor variations in the gonadal activity. Since partial effect was noted in the double-implant group, failure to suppress gonadal activity in these animals is unlikely to be attributed to differences in the hormonal regulation of reproductive cycle which, similarly to other vertebrates, relies on the hypothalamic–pituitary–gonadal axis [38,39]. Further studies with a longer follow-up and/or measurement of deslorelin plasmatic concentration are required to investigate the efficacy in multiple breeding seasons.

## Figures and Tables

**Figure 1 animals-11-00660-f001:**
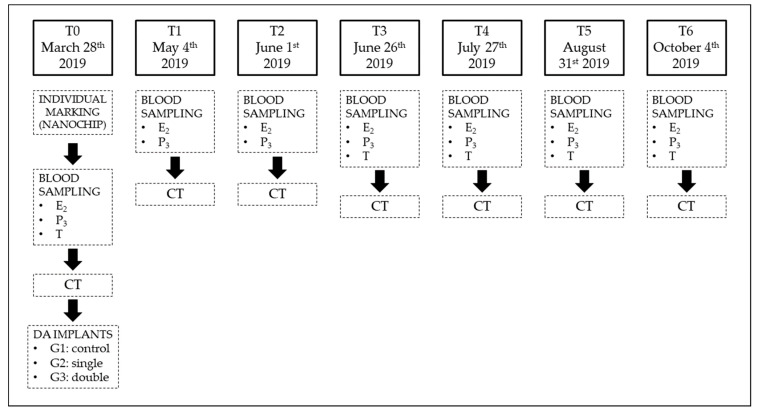
Flowchart illustrating the experimental methodology of the present study. E_2_ = estradiol; P_3_ = progesterone; T = testosterone; G1 = group 1; G2 = group 2; G3 = group 3.

**Figure 2 animals-11-00660-f002:**
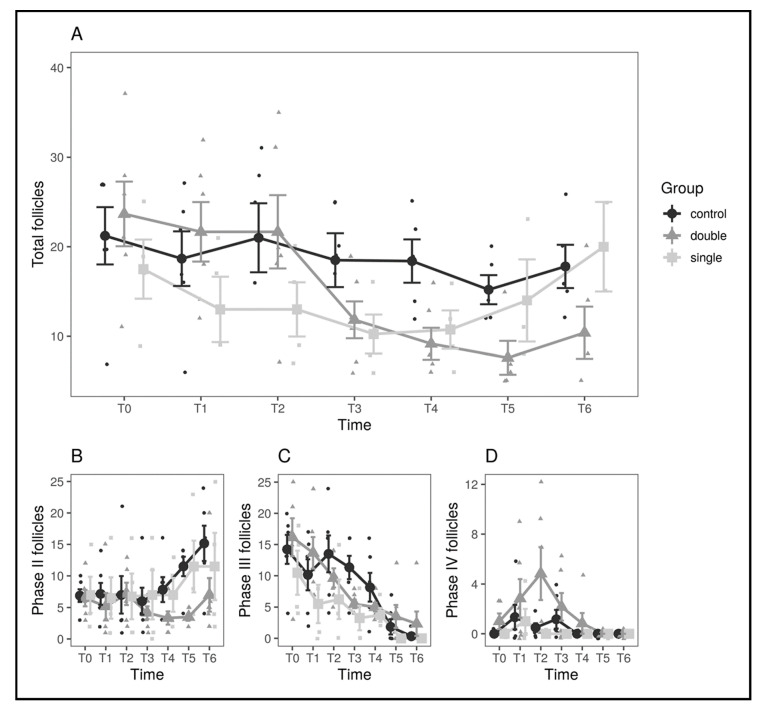
Comparison of the number of follicles between experimental groups and across timepoints. (**A**) Comparison between the control (black), single-implant (light gray), and double-implant (dark gray) groups. (**B**–**D**) Comparison between the three experimental groups for follicles in phases II (**B**), III (**C**), and IV (**D**) separately. In all panels, large dots represent means by group and timepoint, and vertical bars represent standard errors.

**Figure 3 animals-11-00660-f003:**
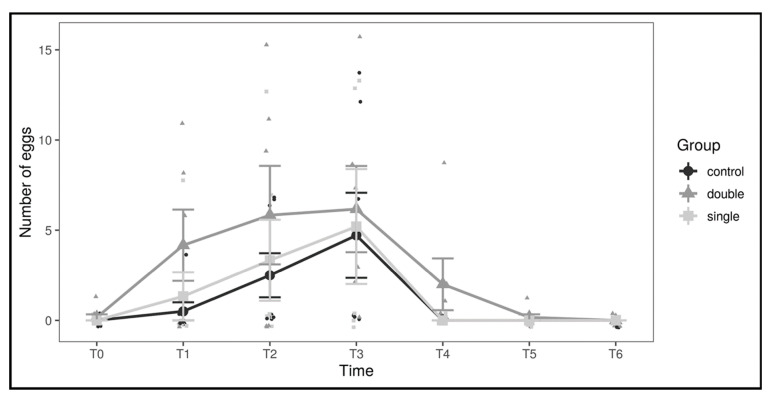
Number of oviductal eggs by experimental group and across timepoints. Large dots represent means by group and timepoint, and vertical bars represent standard errors.

**Figure 4 animals-11-00660-f004:**
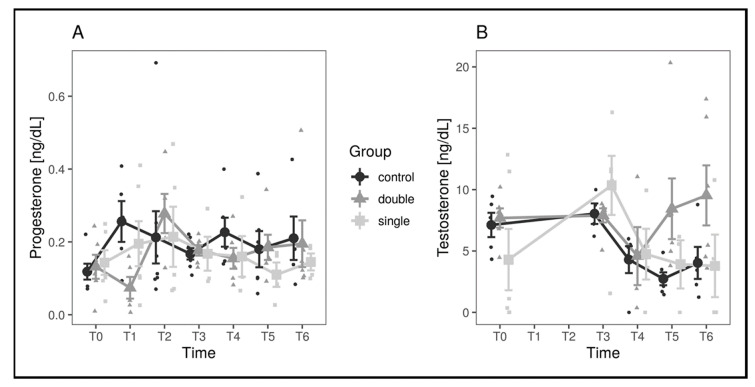
Hormone concentration values in the different experimental groups and across timepoints. Progesterone values are shown in panel (**A**), while testosterone values are shown in panel (**B**). Large dots represent means by group and timepoint, and vertical bars represent standard errors.

**Table 1 animals-11-00660-t001:** Mean, standard deviation (SD), and median values for total numbers of follicles (stages II to IV) and for stages II follicles by experimental group from T0 (28 March 2019) to T6 (4 October 2019).

	Control Group				Single-Implant Group				Double-Implant Group			
	Total Follicles		Stage II Follicles		Total Follicles		Stage II Follicles		Total Follicles		Stage II Follicles	
	Mean (±SD)	Median	Mean (±SD)	Median	Mean (±SD)	Median	Mean (±SD)	Median	Mean (±SD)	Median	Mean (±SD)	Median
T0	19 (±12)	27	5 (±4)	5	16 (±6)	15	7 (±4)	6	24 (±9)	23	6 (±3)	6
T1	17 (±10)	19	6 (±5)	5	15 (±9)	13	8 (±6)	7	22 (±8)	22	5 (±5)	3
T2	18 (±11)	18	6 (±7)	3	13 (±5)	12	6 (±6)	6	22 (±10)	19	7 (±4)	7
T3	15 (±9)	18	5 (±5)	4	8 (±5)	8	6 (±7)	2	12 (±5)	11	4 (±2)	4
T4	12 (±10)	13	8 (±5)	8	7 (±6)	7	7 (±5)	6	9 (±4)	7	3 (±3)	3
T5	10 (±8)	12	11 (±4)	12	8 (±9)	6	11 (±8)	9	7 (±4)	5	3 (±1)	3
T6	12 (±10)	13	15 (±7)	15	8 (±10)	3	11 (±11)	9	9 (±6)	6	7 (±6)	5

**Table 2 animals-11-00660-t002:** Mean, standard deviation, and median values for progesterone and testosterone concentration in all subjects from T0 (28 March 2019) to T6 (4 October 2019).

	Progesterone (ng/dL)		Testosterone (ng/dL)	
	Mean (±SD)	Median	Mean (±SD)	Median
T0	0.13 (±0.07)	0.11	7.17 (±3.53)	7.96
T1	1.82 (±3.68)	0.18		
T2	1.53 (±4.81)	0.22		
T3	0.23 (±0.26)	0.19	8.75 (±3.31)	8.59
T4	0.18 (±0.09)	0.15	4.92 (±3.28)	5.15
T5	0.16 (±0.10)	0,16	5.63 (±4.72)	4.93
T6	0.19 (±0.12)	0.15	7.11 (±5.14)	4.53

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
