# Peer review of "Determination of Efficacy of Single and Double 4.7 mg Deslorelin Acetate Implant on the Reproductive Activity of Female Pond Sliders (Trachemys scripta)"

_animals, 2021, doi:10.3390/ani11030660_

Round 1
Reviewer 1 Report
The manuscript entitled “Determination of efficacy of single and double 4.7 mg desloreline acetate implant on the reproductive activity of female pond sliders (Trachemys scripta)” has been improved according to the reviewer suggestions and now some paragraphs relatives to the treatment dose adopted and the time protocol of animal treatment are clearer for the reader.There are still a number of minor revisions to perform. In the figure 2, the panel A and panel B can be combined in a single panel showing the control and the single and double treatment. Moreover, table 1 should be combined with the table 2 indicating with a single column mean±DS and the second column the median. In the figure 2 as well as in figure 3 the error bars are indicated only with a vertical line without the little horizontal line.
Author Response
Dear Reviewer,
we once again thank you for your time and suggestions. We modiefied the tables and images according to your inputs.
As you will see, there are some revisions in the M&M, Results and Discussion sections made to comply with the other Reviwer's suggestions (basically, he asked to shorten the manuscript and to review some of the Discussion).
Please find attached the new version of the manuscript and a supplementary file with a comparison between the older and newer version.
Reviewer 2 Report
Despite the authors improved the discussion part, the length of the articles did not reduce as it was suggested, moreover, the text slightly enlarged (14 pages vs 13 before). The experiment quite poorly describes the hormonal profile of the slide turtles females. The only important information is the poor effect of the implant on turtles fertility control.
Material and methods could be more compacted, the experimental design could be described in the first paragraph avoiding the redundancy like this: “From T0 (March 28th 2019, before DA implantation) to T6 (October 4th 2019)”.
Line 260-263 – at first describe how looks these seasonal variations, then provide the citations with similar data.
Line 263 – 276 – unnecessary discussion related to the methodology. Not related to the results and its meaning.
The overall discussion part is still poorly written. Authors should go follow the results and discuss each of them in the order reflecting its presence in the Results section.
Author Response
Dear Reviewer,
We once again thank you for the time you dedicated to our work and four your kind suggestions.
Unfortunately, we believe we were not able to fully comply with all your inputs:
- We shortened the manuscript down to 12 pages
- R: Material and methods could be more compacted, the experimental design could be described in the first paragraph avoiding the redundancy like this: “From T0 (March 28th 2019, before DA implantation) to T6 (October 4th 2019)”.
A: Please see new version of M&M in the revised manuscript
- R: Line 260-263 – at first describe how looks these seasonal variations, then provide the citations with similar data.
A: Please see lines 230-233
- R: Line 263 – 276 – unnecessary discussion related to the methodology. Not related to the results and its meaning.
A: We eliminated the section.
- R: The experiment quite poorly describes the hormonal profile of the slide turtles females.
A: we agree with you. It is, however, not the aim of the study to do so: in our work, hormonal evaluation was only a tool to try and investigate the hypothetical efficacy of the drug, and we do not feel adequate to extensively discuss this side topic on the basis of only 8 subjects in the control group. Moreover, it would defy our attempt to “keep it short”.
- We have some difficulty understanding what you mean with “Authors should go follow the results and discuss each of them in the order reflecting its presence in the Results section”. The Results section is organized as follows: 1) results of total follicles count --> 2) results of differential follicle count --> 3) results of hormonal evaluation --> 4) results of correlation between hormones and follicles. We feel that the Discussion follows the same scheme, as we comment (1) and (2) in lines 219-236, (3) in lines 237-249 and (4) briefly in lines 249-252. We could benefit from more specific indications, if you strongly believe that Discussion has to be rewritten.
Please see attached both the revised manuscript and a document comparing the older and newer version.
Round 2
Reviewer 2 Report
The article was improved. Despite this, some minor comments are included in the pdf file.

Author Response
Dear Reviewer,
thank you so much for your patience and your contributions, which have allowed us to improve the quality of our work.
We took care of your final comments: regarding the second one, we did not want to imply that only the november evaluation would have been stressful, but that we tought it was excessive to expose the animals to transportation and wide thermal variations in the days immediately preceding hibernation. We eliminated the last sentence of the paragraph to avoid misunderstanding.
This manuscript is a resubmission of an earlier submission. The following is a list of the peer review reports and author responses from that submission.
Round 1
Reviewer 1 Report
The manuscript entitled “Determination of efficacy of single and double 4.7 mg desloreline acetate implant on the reproductive activity of female pond sliders (Trachemys scripta)” aims to evaluate the efficacy of a single or double intramuscular deslorelin acetate implants during a one-season follow up by examining plasmatic sexual hormones concentration and morphometric evaluation of ovarian. The topic is interesting considering that various aspects of the reproductive function in these animal specie are still unknown.
The manuscript albeit well-conceived shows different methodological flaws. Moreover, the data are not clearly shown. In addition, considering the small number of animals enrolled in the study and the loss of determinations during the experimental protocol, the authors should point more in the discussion.
In particular, the manuscript should be extensively modified.
- In the introduction the authors should be better explain the choice of a single and double implant for DA.
- In the description of animal groups, the treatment, and the time point for the morphometric and hormonal evaluation, it should be more appropriate and clearer for the readers to add a scheme illustrating all these procedures.
- In the description of the methodology of plasmatic hormonal concentration the authors should add the assay sensitivity and then the intra and inter-assay variability.
- In the statistical paragraph the authors should add more detail on the expression of results (medium ±SD) the condition of significative results.
- The quality of the figures is poor and lacking in clarity
- The authors should add more data relative to possible correlations between morphometric evaluation and hormone parameters.
- In the discussion the authors should detail the limits of the study and better discuss the obtained results
- On these bases, the manuscript requires major revisions.
Author Response
Dear Reviewer,
we thank you for the time you dedicated to our work and we appreciate your kind suggestion and comments on how to improve the quality of the present paper.
- R: In the introduction the authors should be better explain the choice of a single and double implant for DA.
- A: Please see lines 80-82 of the revised manuscript
- R: In the description of animal groups, the treatment, and the time point for the morphometric and hormonal evaluation, it should be more appropriate and clearer for the readers to add a scheme illustrating all these procedures.
- A: To try and comply with the reviewer’s suggestion and to make the experimental protocol clearer, we added a flowchart: please see figure 1 of the revised manuscript (line 146)
- R: In the description of the methodology of plasmatic hormonal concentration the authors should add the assay sensitivity and then the intra and inter-assay variability.
- A: Please see lines 117-120 of the revised manuscript
- R: In the statistical paragraph the authors should add more detail on the expression of results (medium ±SD) the condition of significative results.
- A: We agree with the reviewer’s opinion that the results should be better expressed. In order not to make the Results section excessively “heavy” (as, in our opinion, it would be if we wrote a long series of numbers in the main text), we resorted to report the obtained values in tables (Tables 1 to 3).
- R: The quality of the figures is poor and lacking in clarity
- A: We had some difficulties understanding what the reviewer intended here. We modified the graphs making the lines thinner in order to have a clearer view of each group’s variations, but if the reviewer thinks the images still need modifications, we would kindly ask for more specific suggestions.
- R: The authors should add more data relative to possible correlations between morphometric evaluation and hormone parameters.
- A: Please see lines 219-223 of the revised manuscript
- R: In the discussion the authors should detail the limits of the study and better discuss the obtained results
- A: We modified and shortened the Discussion section to try and comply with both Reviewer 1 and 2’s suggestions.
Reviewer 2 Report
The article entitled „Determination of efficacy of single and double 4.7 mg deslorelin acetate implant on the reproductive activity of female pond sliders (Trachemys scripta)” elaborate the use of GnRH-agonists to induce infertility in the invasive reptile species.
Despite the results shows the little effect and rather in the long-term, shows some new information regarding reproduction control in this species.
Despite the needs and necessity of this work is well stated, the methodology is correct, the outcomes are yet scarce.
Even those, due to lack of solid scientific information regarding the topic, this report is worth to publish however in quite shorter form.
Also, the discussion needs to be improved and must be “results-based” dialogue with other authors, rather than a description of the reproductive biology of turtles.
Overall, I recommend this article for publication after shortening the length and revising the discussion part deeply.
The specific comments are given in the text.

Author Response
Dear Reviewer,
we thank you for the time you dedicated to our work and we appreciate your kind suggestion and comments on how to improve the quality of the present paper.
You will notice some changes in the Materials & Methods and in the Results sections (additional figure and tables) to comply with the other reviwer’s suggestions.
We appreciated your comments regarding the Discussion (it was indeed rather pedantic!) and modified the section, trying to better discuss our results both “alone” and in the context of published literature, which is alas scarce.
Regarding the length of the paper, we shortened it from 3075 to 2716 words (main text only, tables and figure captions excluded).